



# Estimating soil organic carbon stocks in *Pinus halepensis* mill. stands using lidar data and field inventory

David Moreno-Pérez[1], María-Belén Turrión[1], Felipe Bravo[2], Irene Ruano[2], Celia Herrero de Aza[1], Frederico Tupinambá-Simões[2]

[1] Department of Soil Science and Agricultural Chemistry. University Institute for Research in Sustainable Forest Management (iuFOR), Escuela Técnica Superior de Ingenierías Agrarias de (ETSIIAA). University of Valladolid, Avda. Madrid, 57. 34004. Palencia, SPAIN.
[2] Department of Plant Production and Forest Resources. University Institute for Research in Sustainable Forest Management (iuFOR), Escuela Técnica Superior de Ingenierías Agrarias de (ETSIIAA). University of Valladolid, Avda. Madrid, 57. 34004. Palencia, SPAIN.

*Correspondence to*: David Moreno-Pérez (dmorper@uva.es)

**Abstract.** Accurate estimation of soil organic carbon (SOC) in forest ecosystems is essential for quantifying their contribution as carbon sinks and improving management strategies in the face of climate change. The objective of this study was to model SOC in *Pinus halepensis* Mill. stands using structural metrics derived from LiDAR data from the National Aerial Orthophotography Plan (PNOA). The study area covered 46.8 hectares located in the municipality of Ampudia, Palencia (Spain). To carry out the work, systematic soil sampling and a forest inventory were conducted. LiDAR technology was also applied and 87 structural metrics were obtained. These metrics were integrated with edaphic variables and above-ground biomass data to build predictive models of carbon stock using multivariate regression techniques.

Among the models evaluated, the Random Forest algorithm showed the best performance in cross-validation ($R^2$ = 0.81; RMSE = 7.73 Mg/ha), demonstrating adequate predictive capacity compared to other models. The proposed approach made it possible to evaluate the potential of LiDAR data from airborne laser scanning (ALS), acquired within the framework of general mapping programmes, as an effective tool for the spatial estimation of SOC. This procedure, validated on an empirical basis, provides a useful methodological basis for advancing in the estimation of SOC through remote sensing, contributing to improve the quantification of soil-related ecosystem services.

## 1. Introduction

### 1.1 Soil monitoring act

Soil monitoring has become a central focus of the European Union environmental strategies, with the aim of ensuring the sustainability of terrestrial ecosystems and mitigating the effects of climate change (Panagos et al., 2020). Quantifying soil carbon is a basic prerequisite for assessing the role of forest ecosystems in mitigating climate change, particularly in the context of climate policies such as the EU Soil Protection Strategy for 2030 (European Commission, 2021). The recent legislative proposal for the Soil Monitoring Act, presented by the European Commission on 5 July 2023, establishes a





comprehensive framework for the assessment, conservation and sustainable management of soils in all Member States, with the aim of achieving the soil status by 2050 (European Comission, 2023). The carbon market is an important tool for meeting global climate goals in the short and medium term by promoting emissions reductions, carbon offsets, and investment in
technologies to reduce emissions (Jiao et al., 2023). However, accurate estimation of soil carbon remains a challenge due to the high spatial and temporal heterogeneity, as well as the variability introduced by different measurement methods (Will, 2017).

In order to ensure transparency and scientific credibility in these mechanisms, especially in land-based carbon initiatives, it
is necessary to establish harmonized monitoring systems. In this line, current European legislation proposes the implementation of a harmonised soil monitoring system, with standardised methodologies for assessing soil quality and ecological functionality (European Comission, 2023). In this policy and scientific framework the present study gets relevance, contributing with empirical evidence on soil carbon estimation under Mediterranean forest conditions. This law represents a socioeconomic opportunity for the agroforestry sector as a whole.

**1.2 Importance of soil organic carbon (SOC) fixation in forest ecosystems**

Soil is an essential component in the provision of ecosystem services, playing an important role in soil-atmosphere interactions, facilitating basic processes such as photosynthesis, improving the water cycle and contributing carbon fixation (Silva & Lambers, 2020; Oishy et al., 2025). SOC fixation and accumulation result from the balance between organic matter input (mainly from litter, fine roots and root exudates) and microbial decomposition levels, which are conditioned by soil,
climate and forest management factors (Lal, 2005; Schmidt et al., 2011).

The capture of $CO_2$ by forest ecosystems through photosynthesis is essential both for net primary production and for reducing the effects of climate change (Navarro Cerrillo et al., 2018). Numerous authors have studied global forest carbon stocks. So, Pan et al. (2011) estimated that global forest carbon stocks were 861 Pg C, with soil up to 1 m deep being the
main reserve (44%), followed by biomass (42%), dead wood (8%) and litter (5%). Carbon fixation is distributed throughout the different ecosystems (Mayer et al., 2020) (Figure 1). There is marked variability in the vertical distribution of organic carbon, with greater accumulation in soils of humid boreal ones in contrast to tropical biotypes, where the dominant fraction is found in above-ground biomass. This heterogeneity highlights the importance of considering the edaphic component in the global carbon balance, particularly in regions where soil is the main reservoir of organic carbon.


The magnitude of carbon stocks in forest soil is affected by the interaction of factors that regulate site fertility and productivity, such as climate, vegetation, topography, chemical, physical and biological properties of the soil, and parent material, in addition to land use and management practices (Mayer et al., 2020). Many researchers have highlighted that soil carbon stocks can be influenced by factors such as low water retention capacity, the presence of coarse materials and limited





effective soil depth, reinforcing the need to implement forest management strategies adapted to these conditions (Ruiz-
Peinado et al., 2017).

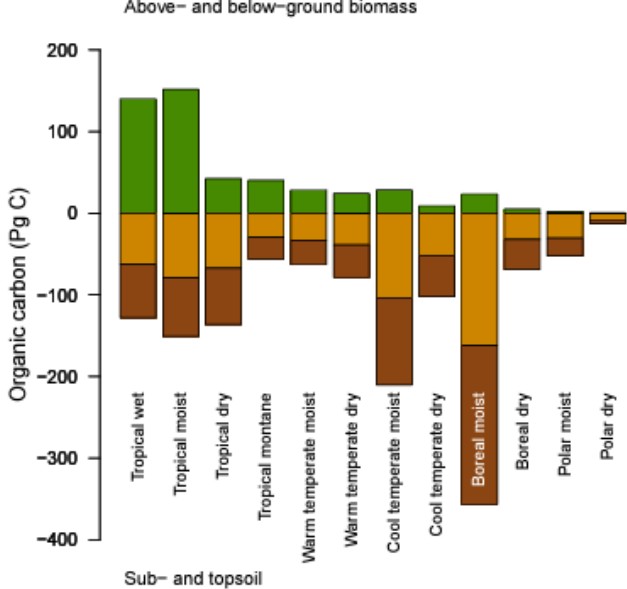

**Figure 1. Organic carbon stocks in the subsoil (brown) and topsoil (orange), and in above-ground and below-ground biomass**
**(green). Figure from Mayer et al. (2020) modified from Scharlemann et al. (2014).**

Therefore, it is increasingly important to establish assessment parameters to estimate soil health, both at the local level,
enabling farmers and forest owners to analyse the impact of their management practices, and at the national level, so that
countries can provide more information on the state of their soil resources (Rabot et al., 2024). In this context, soil organic
carbon (SOC) is one of the most important soil elements, as it directly influences plant growth and essential nutrient cycles
(Navarrete-Poyatos et al., 2019).

In the Mediterranean context, where soil resilience to degradation processes is particularly limited, the role of SOC becomes
even more important. Its conservation contributes to regulating the carbon balance and maintaining ecosystem functions such
as fertility, water retention and soil structural stability (Muñoz-Rojas et al., 2015). In this type of ecosystem, the capacity of
forest soils to store carbon is particularly relevant, given that these are highly vulnerable to aridity, climate variability and
historical land use degradation (Chevallier et al., 2016). Consequently, an accurate and particularly explicit estimate of SOC
content is important for understanding the underlying ecological processes and designing effective forest policies aimed at
sustainable management and adaptation to climate change.




### 1.3 Relevance of *Pinus halepensis* Mill. in carbon storage in the Mediterranean region

*Pinus halepensis*, commonly known as Aleppo pine, is a species native to the Mediterranean region, extending from the Iberian Peninsula to the eastern Mediterranean, including countries such as Spain, France, Italy, Greece and North Africa. Its distribution covers approximately 3.5 million hectares, concentrated mainly in the western Mediterranean basin, especially in Spain and southern France (Figure 2).

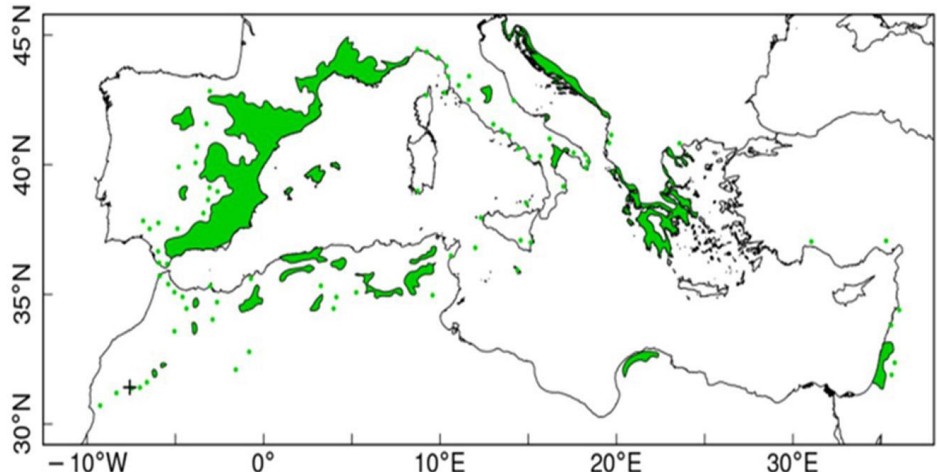

**Figure 2. Native range of Pinus halepensis Mill. in the Mediterranean region.**
**Adapted from Vieira et al. (2022), as cited in Alsanousi et al. (2025).**

Climate change is altering the dynamics of Mediterranean forests, affecting their carbon storage capacity and increasing the risk of desertification (del Río et al., 2008).

Its adaptability to semi-arid conditions and poor soils allows it to colonise degraded areas, contributing to the restoration of disturbed environments and the provision of ecosystem services (Derak & Cortina, 2014; Alsanousi et al., 2025). Thanks to its high ecological plasticity, this species acts as an early coloniser in disturbed ecosystems, being able to regenerate efficiently on bare soils with relatively poor hydrological conditions (Serrada et al., 2008). In turn, *Pinus halepensis* has proven to be an efficient carbon sink (Ruíz-Navarro et al., 2009; López-Senespleda et al., 2021; Santonja et al., 2022). The conversion of agricultural land to forest through reforestation with *Pinus halepensis* can generate significant carbon sequestration over several decades (Charro et al., 2008). The implementation of silvicultural practices, such as systematic





clear-cutting and selective thinning, have been shown to influence growth, carbon storage and sequestration in Aleppo pine reforestations in central Spain (De las Heras et al., 2012; Lull et al., 2024).


### 1.4 LiDAR as a tool for predicting SOC in forest ecosystems

LiDAR (Light Detection and Ranging) is an active laser technology widely used in forest studies (Dassot et al., 2011; Borsah et al., 2023) due to its ability to accurately characterise canopy structure and estimate variables such as biomass or tree height (Oehmcke et al., 2021). LiDAR technology allows us to characterize the three-dimensional characterisation of forest

canopy by emitting and recording laser pulses, generating metrics that describe the vertical distribution of vegetation, canopy density, average or maximum tree height, terrain roughness and other attributes associated with biomass (Lefsky et al., 2002; Tupinambá-Simões et al., 2025). Currently, advances in remote sensing techniques have allowed field inventory to be replaced and/or complemented by airborne laser scanning (ALS) (Navarro Cerrillo et al., 2018). Although significant progress has been made in estimating global soil carbon stocks, the high spatial and temporal variability of carbon reserves

hinders uniform and accurate assessment (Wang et al., 2021; Usman & Begum., 2023; Aroca-Fernandez et al., 2025). In this context, ALS datasets from initiatives such as the National Orthophotography Plan (PNOA) offers an innovative opportunity to improve traditional estimates of soil carbon storage. LiDAR technology allows us to obtain forest structure, reducing uncertainty in the quantification of SOC.

Numerous authors have worked on modelling forest structural variables using ALS LiDAR data and algorithms such as k-nearest neighbours (KNN) or Random Forest (RF) (Yavari & Sohrabi, 2019; Adhikari et al., 2023; Pereira et al., 2023; Strunk & McGaughey, 2023). However, there are not many studies about modelling SOC using LiDAR metrics. Rasel et al. (2017) estimated variables obtained using ALS LiDAR (elevation, forest type and above-ground biomass) to carry out modelling that predicted SOC reserves using the Random Forest (RF) algorithm. Navarro Cerrillo et al. (2018) designed a

methodology to facilitate silvicultural decision-making in forest management by estimating SOC and other variables such as above-ground biomass in stands of *Pinus halepensis* in southern Spain using the KNN algorithm. Moreno Muñoz et al. (2024) developed a model using the RF algorithm to estimate organic carbon storage of mangrove ecosystems in the Colombia southern Pacific coast, demonstrating the potential of machine learning techniques to predict edaphic variables in tropical coastal environments.


Moreover, several studies in soil have demonstrated high accuracy in estimating soil properties, such as SOC and total nitrogen (TN), using high-resolution digital elevation models (DEMs) specifically generated by LiDAR technology (Zhou et al., 2020; Zou et al., 2024). In addition, environmental variables derived from site variables have been used to improve predictions (Mendes & Sommer, 2023). Spatial and temporal dynamics of carbon have been analyzed at regional level, such

as Farina et al. (2017), who proposed studies based on biophysical models (RothC10N) linked to GIS combined with a





database of soil, land use and climate, or Reddy et al. (2015), who carried out an estimate of the extent of soil carbon loss after a forest fire and uncertainty. On the other hand, high-resolution multispectral sensors have already been put into orbit, such as Sentinel-2 (S2), whose ability to quantify SOC content is comparable to that of future hyperspectral space sensors (Castaldi et al., 2019). Topography-based Relief Prediction and Classification Systems (RPCS) offer advantages when

simulating soil redistribution and associated SOC dynamics. Topographic information can be easily obtained from DEMs. The recent increase in the accessibility of high-spatial-resolution LiDAR data can help improve the accuracy of landscape topography derived from DEMs and benefit research in regions with data use constraints. Topography-based models can effectively quantify soil redistribution and SOC distribution patterns (Li & McCarty, 2018). Therefore, one of the main challenges in determining SOC distribution is significant spatial variability, along with the uncertainty it generates (Will,

150 2017).

This context highlights the importance of continuing to optimise methodologies for quantifying and monitoring SOC, which underlines the relevance of this study in the use of advanced technologies such as LiDAR to improve the accuracy of carbon estimation in forest soils. Integrating LiDAR-derived structural data with soil and biomass variables requires addressing

issues such as scale mismatches, spatial resolution, and model generalisability, particularly in heterogeneous Mediterranean landscapes. The findings of this study may inform forest managers and policymakers in designing evidence-based strategies for soil carbon conservation and climate mitigation under Mediterranean conditions.

### 1.5 Objectives of the study

The main objective of this study is to model the SOC content in *Pinus halepensis* forests by integrating structural variables

derived from airborne LiDAR data with soil and forest inventory data. The aim is to generate SOC estimates applicable to forest management in Mediterranean environments, contributing to the development of more accurate and efficient methodologies for soil carbon monitoring. This objective aligns with the EU goalof achieving reliable SOC stimates, a key indicator for assessing forest health.

## 2. Materials and methods

### 2.1 Description of the study area

The study area is located in Ampudia, Palencia (Spain) in the north of the Iberian Peninsula (41º 51'48'' N; 4º 46'13'' W) (Figure 3). The area, situated at 860 meters above sea level (masl), shows a sub-Mediterranean continental climate (Dsb) (AEMET, 2024). The average temperature in the study area is 11.3 ºC and the average annual rainfall is 393 mm for the time series between 1995 and 2024. The present study is sited in *Pinus halepensis* afforested stand (60 years old), with sporadic

presence of *Quercus faginea* Lam*.,* which is sited in the lower stratum and clearly subordinate in terms of coverage and dominance. Table 1 shows the main dendrometric and forest structural variables of the study area (Bueis Mellado, 2017).



**Table 1. Main dendrometric and dasometric variables obtained through the forest inventory.**

| N (trees/ha) | Ht (m) | Ho (m) | Dbh (cm) | G (m²/ha) | — |
|---|---|---|---|---|---|
| 845 | 8.5 | 10.12 | 17.73 | 23.20 | |


Note: N = number of trees per hectare; Ht = average total height; Ho = dominant height; Dbh= diameter at breast height; G = basal area

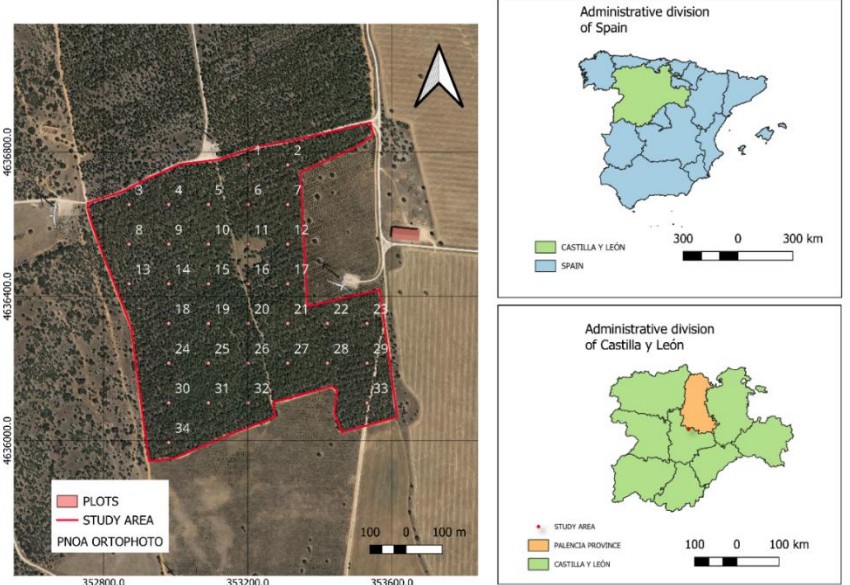

**Figure 3. Location and spatial distribution of the study plots within the experimental study area.            Note: the plots are overlaid on a PNOA orthophoto using UTM Zone 30N projection and ETRS89 coordinate system.**

### 2.2 General methodology

Different data sources (soil data, forest inventory data and LiDAR metrics) were used to model SOC (Figure 4). Soil
sampling data parameters were determinated to estimate soil carbon stock. The forest inventory provided dendrometric variables to estimate tree biomass and C using allometric equations. Finally, processed LiDAR dataset from the PNOA were used to determine structural stand variables.





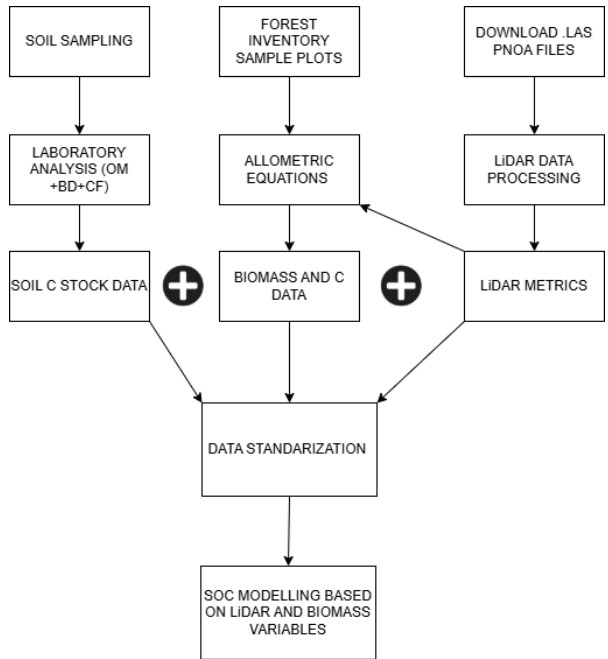

**Figure 4. Road map of the methodology used in this study.**

## 2.3 Soil sampling and laboratory analyses

34 plots were established to carry out the soil sampling. A systematic sampling design was carried out in the whole study area (46.8 ha). All plots were georeferenced using high-precision sub-meter GPS, ensuring the replicability of the sampling design and enabling its accurate integration with the other dataset (forest and LiDAR metrics).

Mineral soil samples were taken in each plot at depth of 0-10 cm. Each plot was divided into four parts, from 6 to 8 individual randomized samples were taken and composited into one homogenized sample per plot. Samples were transported to the laboratory. Mineral soil samples were air-dried, sieved and the percentage of coarse soil materials ([Ø > 2 mm]) was calculated. In each sample, bulk density was measured through the core method (Blake & Hartge, 1986) in the field with volumetric steel rings and soil dry weight.

Soil organic matter was determinated by Walkley and Black method (de Vos et al., 2007), based on the partial oxidation of organic carbon with potassium dichromate in an acidic medium. After the reaction, the excess dichromate not consumed was titrated with ferrous ammonium sulphate (Mohr's salt) in the presence of diphenylamine as an indicator. The percentage of easily oxidizable organic carbon was calculated from the volume of this compound used and the dry weight of the sample. To determine the total carbon in the soil, the total organic matter was divided by 1.724 (MAPA, 1994) [Eq. (1)].



$$SOC = \frac{SOM}{1.724} \tag{1}$$

Accumulated $SOC_i$ (Mg C ha$^{-1}$) [Eq. (2)] was calculated considering C concentration, bulk density, thickness and the percentage of gravels (MAPA, 1994) using the following equation (Lee et al., 2009):

$$SOCi = OCi * BD * (1 - CFi) * ti * 100 \tag{2}$$

Note: $SOC_i$ is the stock of C (Mg C ha$^{-1}$) at depth i; $OC_i$ is the organic C content of the fine soil fraction at depth I; $BD_i$ is the bulk density at depth i (Mg/m$^3$); $CF_i$ is the volumetric content of the coarse fraction at depth i (%) and t is the horizon thickness (m).

All the soil analysis was carried out at the soil laboratory of the ETSIIAA (University of Valladolid).

## 2.4 Forest inventory and generation of equations for quantifying biomass and Carbon

Diameter at breast height (Dbh, with CODIMEX L manual caliper) and total height (Ht, with high-precision Laser GEO Vertex) were recorded in 250 selected trees (Dissanayake, 2024) to develop a site-specific height – diameter equation. Multiple functional forms were tested (linear, exponential, logarithmic, polynomial and potential models) using SAS software with the aim of identifying the best-fit relationship between Dbh and Ht for the study area (SAS Institute Inc., 2023). The selected equation was subsequently used to estimate tree biomass using species-specific allometric models for *Pinus halepensis* (Ruiz-Peinado et al., 2011). So, the following tree biomass components were obtained: Ws: stem with bark (commercial volume, up to a top diameter of 7 cm), Wmb: medium branches (diameter between 2 and 7 cm), Wthinb: thin branches (diameter smaller than 2 cm) and Wr: coarse roots (Table 2). Aboveground biomass (Wa) was defined as the sum of the aboveground biomass fractions of all live trees (Ws, Wmb, Wthinb). Tree carbon (C) fixation was determined by multiplying each biomass value by a generic C concentration of 50.0 %, according to Kollmann (1959) and the Intergovernmental Panel on Climate Change (IPCC) recommendations (Penman et al. 2003).




**Table 2. Equations for estimating biomass for Pinus halepensis (Ruiz-Peinado et al., 2011).**

| Variable | Equation |
|---|---|
| Ws | $Ws = 0.0139 * d^2 * h$ |
| Wr | $Wr = (0.0785 * d^2)$ |
| Wb7 | $Wb7 = [3.926 * (d - 27.5)] * Z$ |
| Wb2-7 | $Wb2 - 7 = 4.257 + 0.00506 * d^2 * h - 0.0722 * d * h$ |
| Wb0,5-2 | $Wb0,5 - 2 = 6.197 + 0.00932 * h^2 * 1 - 0.0686 * h$ |

Note: Ws = stem biomass; Wr = root biomass; Wb7 =Biomass weight of the thick branch fraction (diameter larger than 7 cm); Wb 2-7 = biomass of branches from 2 cm to 7 cm; Wb 0.5-2 = biomass of branches from 0.5 cm to 2 cm

**2.5 ALS data from the PNOA and data processing**

For the ALS data, Leica ALS80 was used as the main sensor. Technical specifications of the data, obtained in 2019, are presented in Table 3. The flight over the study area took place in 2019 (Figure 5).



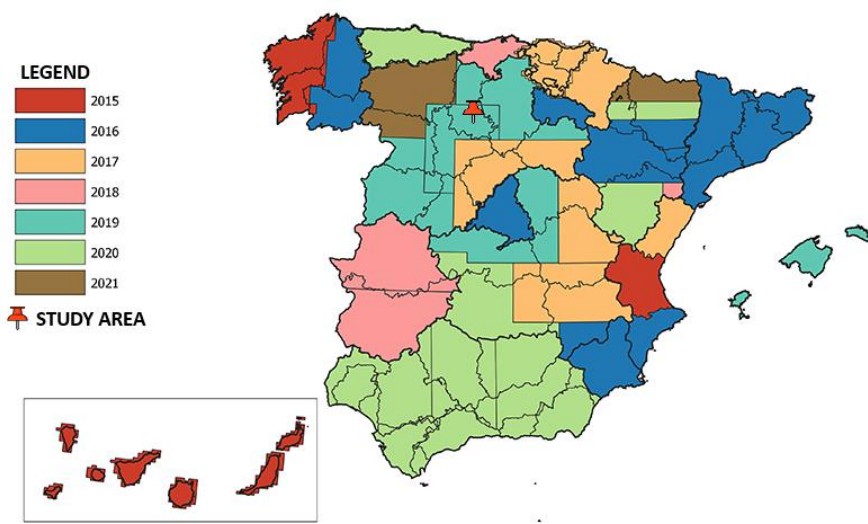


**Figure 5. Year of commencement of the second cycle of the PNOA-LiDAR project by zone. Source: PNOA.**

**Table 3. Technical specifications of the second coverage of the PNOA – LiDAR project. Source: PNOA.**

| | |
|---|---|
| **Minimum point density** | $0.5 - 2$ points/m$^2$ |
| **Year of flight** | 2019 |
| **Geodetic reference system** | ETRS89 huso 30 N |
| **RMSE Z** | $\leq 20$ cm |
| **Estimated planimetric accuracy** | $\leq 30$ cm |
| **Simultaneous image** | Si |
| **File size** | 2 x 2 km |
| **File format** | LAS 1.2 format 3 |
| **MDE grid spacing** | 2 m x 2 m |
| **RMSE Z (MDE)** | $\leq 25$ cm |
| **Estimated planimetric accuracy (MDE)** | $\leq 50$ cm |

Note: RMSE Z = Root mean square error in the Z component; MDE = Digital Elevation Model

LiDAR data was downloaded from the National Geographic Information Centre portal in .LAS format, selecting the files corresponding to the map sheets that completely covered the study area. The .LAS files were then spatially cropped using a previously digitized vector layer that precisely delimited the perimeter of the analyzed area. The LiDAR data was processed at the plot scale, addressing each sampling unit individually. To do this, the R programming environment (version 4.4.1) was





used (R Core Team, 2024), employing a combination of packages specialised in geospatial analysis and point cloud
processing, such as lidR, terra and sf (Pebesma, 2016; Hijmans, 2025). The lidR package provides a wide range of functions
for manipulating and visualising airborne LiDAR data, including reading and writing .LAS and .LAZ files, point
classification, digital terrain and canopy model generation, height normalisation and individual tree segmentation (Roussel et
al., 2020; Peter et al., 2021).

First, the files were read to extract a point cloud that included both the file header information (coordinate reference system
and format type) and the position data and attributes associated with each point in the cloud. Specific attributes were then
selected, such as X, Y, and Z coordinates and intensity, in order to optimize memory management and speed up processing.
In addition, specific attributes and filters were determined to eliminate non-essential data, such as points that did not
correspond to the first return, which increased the efficiency of the analysis by reducing the amount of data to be processed.
Next, a filter was applied to extract vegetation points, considering that points corresponding to the terrain had heights of less
than 2 m. The Digital Terrain Model (DTM) and the Canopy Height Model (CHM) were generated for the entire study area.
Subsequently, the structural metrics of the vegetation were calculated, such as the average height, maximum height and
minimum height in meters. Based on the CHM, the height peaks representing tree crowns were identified, and the CHM was
segmented to delimit or identify individual trees. To generate the DTM, only points classified as terrain were used, applying
the Inverse Distance Weighting (IDW) method. This approach assigns values to unsampled locations using a weighted
average of neighboring points, with weights inversely proportional to the distance raised to a specified power, commonly
used in geospatial studies to interpolate continuous surfaces (Shepard, 1968; Moussa & Abboud, 2024). In this study,
interpolation parameters were defined with k=10 nearest neighbours and a power p=2. Subsequently, the LiDAR point cloud
was normalized by adjusting the heights of non-terrestrial points in relation to the DTM, allowing for a representation of the
vertical structure of the forest canopy. To generate the CHM, the 'pit-free' algorithm was applied, designed to avoid artificial
depressions in canopy models by combining multiple surfaces generated at different height thresholds (Khosravipour et al.,
2014). This CHM was used to identify and characterize individual tree crowns, extracting metrics such as the maximum
height of each tree.





A conceptual map of LiDAR data processing is presented (Figure 6).

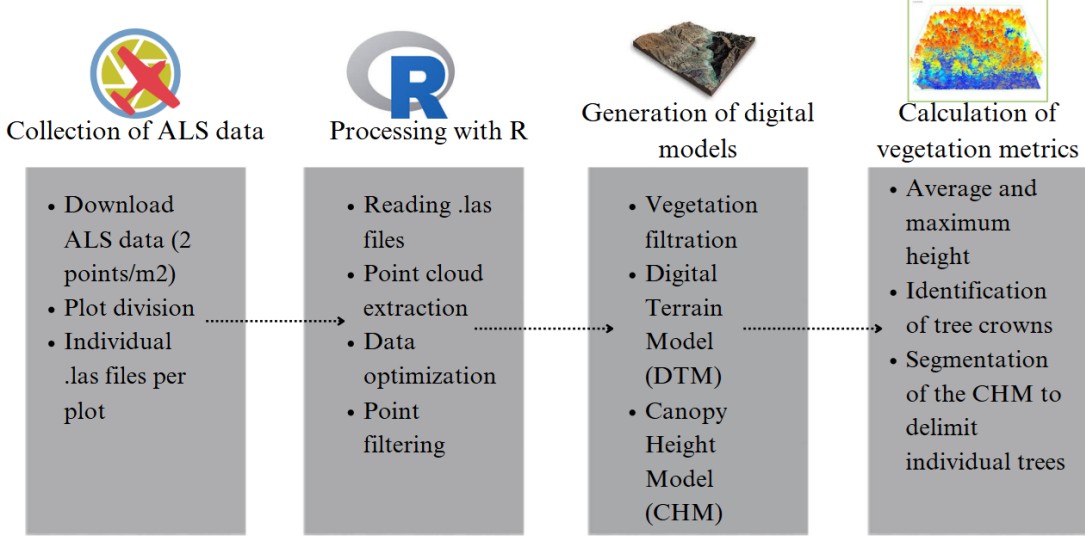

**Figure 6. Concept map of LiDAR data processing.**

Once the point cloud was processed, multiple structural metrics derived from LiDAR returns at the plot level were calculated using the lidar_metrics function from the lidR package (Roussel & Auty, 2022). The variables extracted included descriptive height statistics such as total number of returns, maximum height, minimum height, mean height, standard deviation, coefficient of variation, skewness, and kurtosis (n, zmax, zmin, zmean, zsd, zcv, zskew, zkurt). Height percentiles were also calculated in increments from 1 to 99% (zq1 to zq99), proportions of returns above certain thresholds (e.g., 2 m, 5 m,

zmean), as well as cumulative metrics and proportions within specific height ranges. Likewise, indices derived from L moments (L1 to L4, Lskew, Lkurt), metrics based on the leaf area density profile (lad_min, lad_max, lad_mean, lad_sum, lad_cv) and statistics associated with the position of the return within the pulse (n_first, n_last, n_single, n_multiple, p_first, p_last, etc.). All metrics were calculated individually for each plot and normalised for subsequent use as predictors in the SOC estimation models. For a detailed description of each variable, see Suplement. (Tables S.2. to S.9.).

**2.6 SOC modelling and generation of the predicted SOC map**

The modeling approach consisted of applying different models and machine learning algoritms, with the aim of evaluating the predictive capacity of SOC through soil data, tree biomass variables and LiDAR metrics. So, firstly, Lidar data was processed. Predictor variables with a variance of less than 0.01 were removed as they were considered non-informative. The collinearity between the variables was examined by calculating the Pearson correlation matrix, removing those with



correlation coefficients greater than 0.9 in order to reduce redundancy and prevent multicollinearity problems. The remaining variables were normalized through standardization (mean zero and standard deviation one).

Secondly, the processed dataset was randomly divided into two subsets: a training subset comprising 75% of the observations and a test subset comprising the remaining 25%. This division was performed in a stratified and reproducible

manner, using a fixed random seed. Four regression models with different levels of complexity were defined and implemented: a simple linear model, a second-degree polynomial model, a logarithmic model (log(x+1)) and a Random Forest model. All models were validated using a cross-validation scheme with 10 partitions (k-fold cross validation, v=10) on the training set. In the case of the Random Forest model, hyperparameter optimization was carried out using a random search with 20 different combinations of the parameters mtry (number of predictors considered in each tree division) and

min_n (minimum number of observations in a terminal leaf). The performance of each model was evaluated using the statistical parameters root mean square error (RMSE), mean absolute error (MAE) and coefficient of determination ($R^2$) [Eq. (3), (4) and (5)].

$$MAE = \frac{\sum_{i=1}^{n}|y_i - x_i|}{n} \tag{3}$$


$$RMSE = \sqrt{\frac{\sum_{i=1}^{n}(y_i - x_i)^2}{n}} \tag{4}$$

$$R^2 = 1 - \frac{\sum(y_i - x_i)^2}{\sum(y_i - \mu_y)} \tag{5}$$

The statistical parameters obtained for each model allowed us to identify the best alternative in terms of predictive accuracy.


To generate a spatially continuous map of predicted SOC, a wall-to-wall rasterization approach was implemented using R (version 4.4.1) and the terra package (Hijmans, 2025). First, the complete dataset of predictor variables was structured as a spatial point object using the corresponding UTM coordinates (EPSG:25830). The area of interest was rasterized at a spatial resolution of $20 \times 20$ meters, consistent with the spatial density of the LiDAR-derived

metrics (Trouvé et al., 2023). A multi-band SpatRaster template was constructed, covering the full extent of the



study area. The predictor variables used in the selected model were then rasterized layer by layer over this template. Once rasterized, the trained selected model was applied to the full stack of predictor layers using the predict() function, yielding a continuous surface of SOC values in Mg C/ha. The resulting raster was clipped using a shapefile defining the limits of the forest stand. The file containing the SOC information for each pixel was 325 exported in GeoTIFF format for visualisation and subsequent spatial analysis. Following the modelling and prediction procedure, the resulting SOC map was laid out using QGIS (version 3.40).

## 3. Results

### 3.1 Results of soil data

Carbon concentration ranged from 0.63% to 5.08%, while TOM ranged between 1.08% and 8.74%. Higher SOC values (Mg 330 C/ ha) were associated with higher concentrations of organic carbon (OC%) and total organic matter (TOM%).

Bulk density (BD) ranged between 1.16 and 1.85 g / cm$^3$, while coarse fraction percentage (CF%) showed a broader dispersion, from low values (<1%) in several plots to higher proportions (e.g., 30% in plot 22). The results of the edaphic properties analysed (Table 4) showed that SOC (Mg C / ha) varied among plots, ranging from 9.49 Mg C ha$^{-1}$ (plot 28) to 335 53.16 Mg C ha$^{-1}$ (plot 2).

**Table 4. Laboratory results of soil properties per plot: soil organic carbon (SOC0–10), total organic carbon (TOC), total organic matter (TOM), bulk density (BD) and percentage of coarse elements.**

| Plot | SOC$_{0-10}$ (Mg C / ha) | OC (%) | TOM (%) | BD (g/cm3) | CF (%) |
|---|---|---|---|---|---|
| 1 | 14.60 | 1.36 | 2.34 | 1.24 | 13.64 |
| 2 | 53.16 | 5.08 | 8.74 | 1.21 | 13.67 |
| 3 | 16.75 | 1.31 | 2.25 | 1.56 | 17.72 |
| 4 | 38.48 | 1.97 | 3.39 | 1.99 | 1.58 |
| 5 | 19.20 | 2.31 | 3.97 | 1.05 | 20.68 |
| 6 | 19.83 | 1.36 | 2.34 | 1.48 | 1.52 |
| 7 | 18.83 | 1.51 | 2.60 | 1.57 | 20.50 |
| 8 | 17.21 | 1.02 | 1.76 | 1.69 | 0.51 |
| 9 | 16.04 | 1.01 | 1.74 | 1.59 | 0.60 |
| 10 | 20.71 | 1.25 | 2.15 | 1.66 | 0.29 |
| 11 | 18.13 | 1.46 | 2.51 | 1.32 | 5.63 |
| 12 | 30.82 | 1.69 | 2.91 | 1.85 | 1.59 |




**Table 4. (cont.) Laboratory results of soil properties per plot: soil organic carbon (SOC0–10), total organic carbon (TOC), total organic matter (TOM), bulk density (BD) and percentage of coarse elements.**


| Plot | SOC$_{0-10}$ (Mg C / ha) | OC (%) | TOM (%) | BD (g/cm3) | CF (%) |
|---|---|---|---|---|---|
| 13 | 18.59 | 1.21 | 2.08 | 1.54 | 0.42 |
| 14 | 18.70 | 1.37 | 2.36 | 1.36 | 0.22 |
| 15 | 14.50 | 0.80 | 1.38 | 1.82 | 0.16 |
| 16 | 47.25 | 3.51 | 6.03 | 1.52 | 11.41 |
| 17 | 15.96 | 1.02 | 1.76 | 1.56 | 0.22 |
| 18 | 12.43 | 0.91 | 1.57 | 1.57 | 13.26 |
| 19 | 22.15 | 1.68 | 2.90 | 1.45 | 9.15 |
| 20 | 20.15 | 1.44 | 2.48 | 1.45 | 3.39 |
| 21 | 22.62 | 1.45 | 2.50 | 1.58 | 1.64 |
| 22 | 13.72 | 1.49 | 2.56 | 1.39 | 33.68 |
| 23 | 25.67 | 1.56 | 2.69 | 1.67 | 1.33 |
| 24 | 26.08 | 1.81 | 3.11 | 1.51 | 4.37 |
| 25 | 15.38 | 1.03 | 1.76 | 1.53 | 1.92 |
| 26 | 17.02 | 1.32 | 2.27 | 1.44 | 10.50 |
| 27 | 14.53 | 0.94 | 1.62 | 1.57 | 1.67 |
| 28 | 9.49 | 0.63 | 1.08 | 1.51 | 0.51 |
| 29 | 19.11 | 1.69 | 2.90 | 1.29 | 12.07 |
| 30 | 11.20 | 0.71 | 1.22 | 1.60 | 1.60 |
| 31 | 17.48 | 1.23 | 2.12 | 1.52 | 6.55 |
| 32 | 15.41 | 1.64 | 2.81 | 1.16 | 18.83 |
| 33 | 18.44 | 1.45 | 2.49 | 1.45 | 11.96 |
| 34 | 11.58 | 0.78 | 1.35 | 1.52 | 3.12 |

Note: SOC$_{0-10}$ = Carbon Stock from 0-10 cm; OC = organic carbon content of the fine soil fraction (< 2 mm); TOM = Total Organic Matter; BD = bulk density; CF = volumetric content of the coarse fraction.

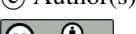



**Table 5. Descriptive statistics of soil variables analyzed in the laboratory (n = 34).**

| | Mean | Standard deviation | Minimum | Maximum | Coefficient of variation (%) |
|---|---|---|---|---|---|
| **$SOC_{0-10}$** | 20.33 | 9.44 | 9.49 | 53.16 | 46.46 |
| **OC (%)** | 1.50 | 0.82 | 0.63 | 5.08 | 54.78 |
| **TOM (%)** | 2.58 | 1.41 | 1.08 | 8.74 | 54.74 |
| **BD (g/cm3)** | 1.51 | 0.19 | 1.05 | 1.99 | 12.63 |
| **CF (%)** | 7.23 | 8.11 | 0.16 | 33.68 | 112.14 |

The average SOC was 20.33 Mg/ha (Table 5), ranging from 9.49 to 53.16 Mg/ha. A similar trend was found in total organic matter values, with an average value of 2.58% and a coefficient of variation of 54.78%. The bulk density ranged from 1.05 to 1.99 g/cm³. In contrast, the percentage of gravels showed the greatest dispersion of all the variables analyzed (CV = 112.10%).

### 3.2 Results of tree biomass estimation

For the h-d model, the linear equation was selected as the best (Table 6) showing a higher $R^2$ (0.546) (Figure 7).

**Table 6. Height diameter models fitted with forest inventory data.**

| Model | Equation | $R^2$ |
|---|---|---|
| Linear | $y = 0.2731x + 4.9865$ | 0.546 |
| Exponential | $y = 6.4408 e^{0.0233x}$ | 0.532 |
| Logarithmic | $y = 6.6774\ln(x) - 9.4012$ | 0.519 |
| Polynomial | $y = -0.001x^2 + 0.3299x + 4.2507$ | 0.535 |
| Potential | $y = 1.837x^{0.5778}$ | 0.542 |



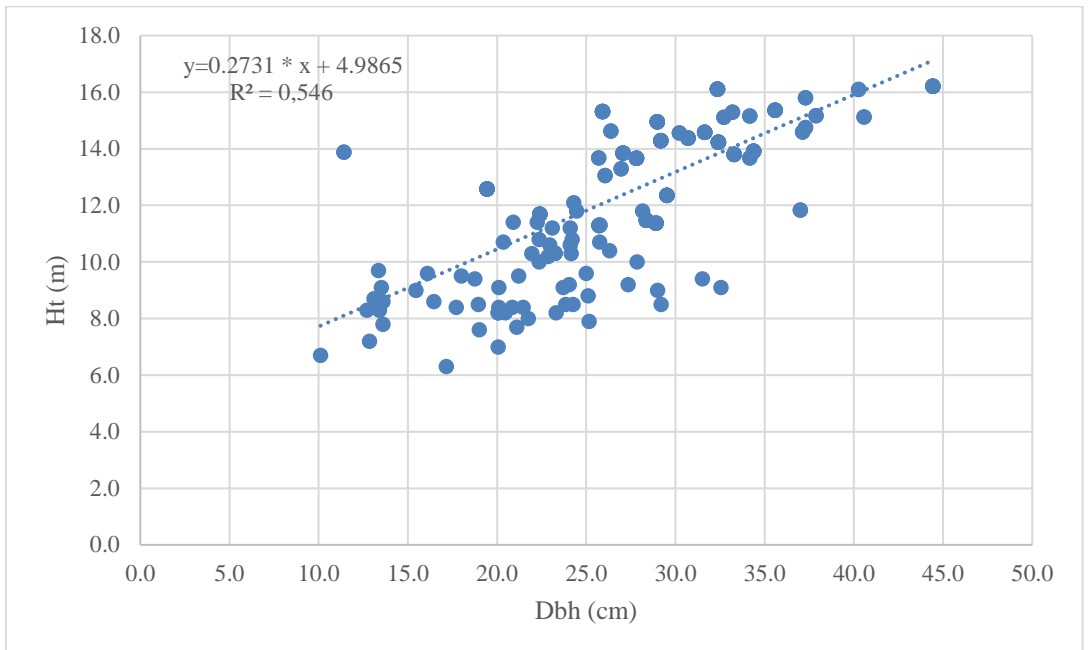

**Figure 7. Fit between height and diameter of measured trees using the linear model.**


$$Ht = 0.2731 * Dbh + 4.9865 \qquad (6)$$

The results of biomass estimation in all plots are shown in Supplement. (Table S.1.).

Stem biomass (Ws) represented the dominant fraction of aboveground biomass across the plots, accounting for an average of 57.23% of the total. It was followed by thick branches (Wb2–7), contributing 27.42%, while small branches (Wb0.5–2)

accounted for 15.35% of the aboveground biomass. Root biomass (Wr), as the belowground compartment, accounted for the total Carbon tree biomass.

**3.3 SOC modelling based on LiDAR and biomass metrics**

Ten-fold cross-validation on the training set showed differential performance between models, with higher performance in the case of Random Forest model (Table 7), far exceeding the results of the parametric models. Given the performance

achieved during cross-validation, the Random Forest model was selected as the optimal candidate for estimating the SOC. Table 7 summarizes the mean values of each metric for the four models evaluated during cross-validation. Figure 8 shows the relationship between observed and predicted SOC values, showing that predictions ranged between 15 and 30 Mg C/ha.




**Table 7. Average performance metrics obtained through cross-validation (10-fold) for soil organic carbon (SOC) estimation models.**

| Model | RMSE | MAE | R² |
|---|---|---|---|
| Random Forest | 7.73 | 6.13 | 0.811 |
| Logarithmic | 331 | 264 | 0.871 |
| Polynomial | 745 | 455 | 0.794 |
| Linear | 661 | 498 | 0.721 |

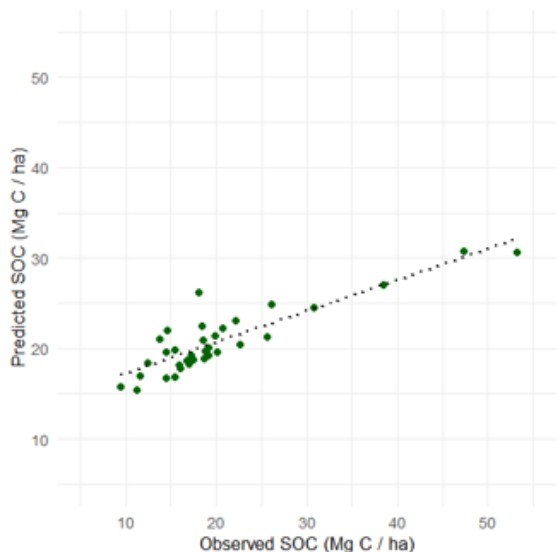

**Figure 8. Random Forest model performance: Predicted vs Observed Soil Organic Carbon (SOC) after cross-validation.**





**3.4 Mapping of the SOC**

SOC content was spatially distributed (Figure 9) and most values ranged between 16.2 and 23.5 Mg C/ha, with a remarkable spatial variability throughout the forest stand.

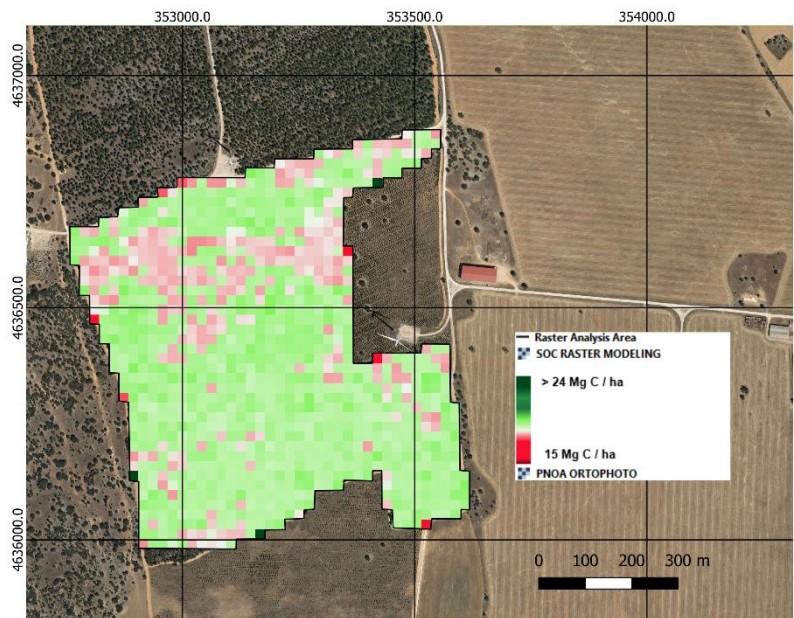

**Figure 9. Predicted spatial distribution of soil organic carbon (Mg C/ha) using the Random Forest model.**

**4.      Discussion**

This study integrated structural data obtained from soil, tree biomass and airborne laser scanning (ALS) to estimate SOC content in a Mediterranean *Pinus halepensis* ecosystem. This approach was based on (a) soil carbon stock data, (b) auxiliary biomass variables calculated from forest inventory (c) structural metrics derived from LiDAR point clouds processed at the plot level, and (d) modelling and machine learning techniques to identify relationships between forest structure and soil carbon contents. The results allowed us to generate relevant predictors of SOC, even under conditions of high vegetation cover, demonstrating the robustness of the modelling framework to operate under structurally heterogeneous forest environments. Our findings emphasize the potential of our approach to be scaled to other semi-arid or Mediterranean ecosystems where field sampling is limited, but airborne LiDAR data is available. Our approach could be applied across broader spatial domains, aligning with the current efforts to develop harmonized soil carbon monitoring strategies at international level.





## 4.1 LiDAR ALS for estimating the SOC

In this study, the application of airborne LiDAR (ALS) data has proven to be effective in capturing structural metrics of forest canopy, providing valuable proxies for estimating SOC in Mediterranean ecosystems dominated by *Pinus halepensis*.
Studies using ALS sensors to estimate forest biomass are common (Breidenbach et al., 2021; Perea-Ardila et al., 2021; Li et al., 2024) and different authors carried out studies of SOC modeling based on satellite images such as Sentinel (Zhou et al., 2020; Amarnath et al., 2024), however there is a lack of studies using ALS sensors to estimate SOC. Our findings showed that metrics derived from ALS can correlate significantly with soil carbon stocks, especially when they were integrated with forest inventory data.


The upper soil layer (0-10 cm) usually concentrates most of the active organic matter and is particularly influenced by processes of carbon accumulation (Hoyle et al., 2011). Previous studies have shown estimations similar to our findings. So, Rasel et al. (2017) fitted a model that estimated 69% of SOC based on biomass variables obtained through ALS sensors and Stumpf et al. (2024), were able to estimate up to 64% of the carbon in the top 1.2 m of soil. Moreover, Navarro-Cerrillo et al.
(2018) found that combining low-density ALS data with nearest neighbor (kNN) models showed 82% of accuracy in carbon stocks in the top 10 cm of soil in *Pinus halepensis* plantations in southeastern Spain. Even Navarrete-Poyatos et al. (2019) identified a significant improvement in the prediction of SOC using ALS data in combination with machine learning models such as Random Forest. Finally, Pascual et al. (2023) addressed the quantification of stored carbon by integrating LiDAR metrics with estimatations of above-ground biomass and litter, highlighting the relevance of incorporating multiple
ecosystem pools, looking for a more complete assessment of carbon content.

Other studies highlighted the usefulness of LiDAR ALS in predicting soil properties. For example, Li et al. (2016) evaluated the effectiveness of LiDAR-derived variables for estimating surface soil horizon properties in a *Pinus koraiensis* forest and showed that LiDAR-derived variables could be predictors of soil properties, with coefficients of determination (R²) ranging
from 0.46 to 0.66. In addition, Hounkpatin et al. (2021) compared global and local models for predicting SOC stocks in Swedish forests, using national forest inventory data and digital soil mapping approaches. Their findings suggested that local calibration has the potential to obtain higher accuracy. Other studies carried out a complex methodology that combined field data with metrics derived from LiDAR ALS to estimate changes in SOC stocks at the stand level, using the Yasso15 model to simulate edaphic carbon dynamics (Strîmbu et al., 2023). In the present study, the Random Forest algorithm was applied
to model SOC from LiDAR metrics obtained from the PNOA. Unlike previous studies that rely on dense sampling networks, detailed spectral data, or mechanistic models like Yasso15, our approach showed that it is possible to achieve accurate SOC predictions using solely airborne LiDAR metrics derived from public datasets. This adds value by confirming the operational potential of low-cost, replicable methodologies based on open-access remote sensing data and machine learning techniques, especially in Mediterranean forest systems where field data are often highly heterogeneous.



## 4.2 LiDAR ALS for estimating the SOC

Despite not having an independent test set evaluation due to sample effort limitations, the robustness of the model in cross-validation allowed these results to be considered as representative of the expected behavior of the model under similar conditions. The Random Forest (RF) model applied in this study showed the best performance among the evaluated approaches for SOC estimation, achieving a coefficient of determination of 0.811 during cross-validation. Beyond its predictive accuracy, this result reinforces the capacity of non-parametric methods to capture nonlinear relationships and interactions between structural LiDAR metrics and soil carbon content. The robustness of RF in handling high-dimensional datasets with potential multicollinearity makes it particularly suitable for heterogeneous Mediterranean forest systems. The associated errors were relatively low, with a mean RMSE of 7.73 Mg/ha and a MAE of 6.13 Mg/ha. In comparison, parametric models (linear, polynomial and logarithmic) had higher errors and lower explanatory power, which limited their predictive utility. These results are consistent with other studies that showed more accurate estimates from machine learning techniques for estimating soil properties by LiDAR data (Navarro Cerrillo et al., 2018; Alonso-Sarria et al., 2025). Rasel et al. (2017) highlighted that RF is particularly effective in capturing nonlinear relationships and managing highly correlated variables, thus outperforming linear regression-based approaches when working with structural proxies derived from remote sensing. Similarly, Misebo et al. (2024) used generalized additive models to estimate SOC in areas restored after mining and obtained significantly lower R² values, demonstrating the advantages of the approach used in this study. However, the absence of certain key edaphic variables should be considered when extrapolating the results to more complex or heterogeneous forest scenarios.

The robustness of the RF model agrees with the findings of Hu et al. (2023), who applied remote sensing-based models and ecological simulations in fire-affected boreal forests, finding that SOC depended on both metrics quality and the suitability of predictive methods. In their study, the combination of field data with remote information allowed models to be calibrated with coefficients of determination greater than 0.80, even under severe disturbance conditions. In addition, Hengl et al. (2017) confirmed the widespread use of Random Forest as a generic framework for modeling complex spatial variables, highlighting its usefulness in predicting edaphic carbon with multisensor data. The performance observed in the present study is within the range of accuracy reported in these studies, reinforcing the applicability of the model in Mediterranean forest contexts with moderate-resolution ALS data coverage.

A tendency of the Random Forest model to underestimate SOC values was observed, particularly in plots with higher stocks values. This systematic underestimation is consistent with findings from previous research. For instance, Agaba (2024) reported that Random Forest models underestimated SOC in heterogeneous mountainous landscapes, especially at high altitudes and in areas with steep slopes. They attributed this pattern to insufficient sampling density in complex terrain and the limited representation of extreme SOC values in the training data. Similarly, Ou et al. (2024) demonstrated that the





inclusion of site attributes and climatic variables significantly improved SOC prediction in cropland soils, highlighting that the absence of such covariates can reduce model performance. Future research could explore whether the inclusion of site or

climatic covariates—absent in the present model—might help reduce the underestimation observed in certain SOC predictions. Similarly, increasing the number of field plots could contribute to improving model robustness and reducing potential bias, particularly in heterogeneous forest environments.

### 4.3 Relationship between forest structure and SOC

Structural parameters such as tree density, tree size, species diversity and aboveground biomass can estimate both

atmospheric carbon fluxes and carbon incorporation and stabilization in the soil through the contribution of litter, roots and exudates (Muñoz-Rojas et al., 2016). In *Pinus halepensis* forests, the relationship between forest structure and SOC is particularly relevant, due to their distribution in many areas of semi-arid Mediterranean ecosystems. However, several studies have shown that *P. halepensis*, a species widely used in reforestation in the Mediterranean area plantations do not always fully restore original SOC levels after disturbances such as fires or previous degradation (Goberna et al., 2007).

These results suggest that, although the above-ground biomass of *P. halepensis* may be considerable, its effect on soil carbon reservoirs could be limited by factors such as the poor quality of the organic matter contributed or accelerated decomposition rates under arid conditions (Lull et al., 2024). This pattern is consistent with the results obtained in the present study, where despite observing plots with high values of biomass and aboveground carbon, no direct and consistent relationship was identified between these values and SOC content. This suggests that carbon input from vegetation does not necessarily

translate into greater edaphic carbon fixation, reinforcing the idea that other factors such as microbial dynamics, textural characteristics or coarse element content may modulate SOC retention and stabilization in the surface soil horizon (Doetterl et al., 2025).

The marked heterogeneity in CF values across plots suggested significant differences in the content of rock fragments or

lithology within the surface soil profile, which may have important implications for water storage capacity, nutrient availability and root development of vegetation. The results obtained showed variability in biomass values between plots associated with differences in stand density, stand structure and forest management intensity. These results highlighted that the structural development plays a key role in aboveground biomass accumulation. Plots with greater structural complexity—such as plots 12, and 9—showed the highest aboveground biomass values, exceeding 80 Mg/ha in the most

developed cases, reflecting advanced stand maturity and significant carbon storage capacity. In contrast, plot 11 with no tree vegetation, pointing to early successional conditions or perhaps disturbance history, showed limited structural development (Hernández-Alonso et al., 2023).



## 4.4 Projections and scalability of the model for monitoring SOC

The integration of remote sensing techniques such as ALS with machine learning-based prediction models has proven to be
an effective strategy for estimating SOC, not only at the local scale, but also with potential for scalability to regional or
international levels. In this study, a set of structural metrics derived from low-density ALS data and forest inventory was
used to model the SOC content in a Mediterranean forest ecosystem with a predominance of *Pinus halepensis.* The results
obtained, especially with the Random Forest model ($R^2$ = 0.811; RMSE = 7.73 Mg/ha), demonstrated the ability of this type
of variable to capture soil patterns with a reasonable level of error.


This approach agrees with studies such as that by Stevens et al. (2013), who integrated site variables derived from digital
terrain models with field data to generate SOC predictions in agricultural soils. Moreover, Brogniez et al. (2015), also used
generalized additive models based on the Land Use/Cover Area frame statistical Survey (LUCAS) database to map SOC in
the surface horizon at a continental scale. Both studies highlighted the usefulness of multivariate approaches that combine
environmental variables derived from remote sensors with limited but representative edaphic information. Our work
contributes to this line of research, demonstrating that, even in scenarios with a small number of plots, the structural
information obtained by ALS can be sufficient to generate robust SOC maps at an operational scale.

In addition, Panagos et al. (2022) highlighted the importance of infrastructures such as the European Soil Data Centre
(ESDAC) for harmonizing data and facilitating its use in soil management policies and climate strategies. Similarly, Ballabio
et al. (2016) demonstrated how harmonized databases such as LUCAS can be used to map high-resolution physical soil
properties using non-linear modelling techniques such as adaptive regression splines (MARS). In the present study, although
with a more restricted scope, similar methodological procedures based on multivariate regression and integration of
topographic and structural variables derived from LiDAR were used to estimate SOC content on a smaller scale. In this
context, the results of our study could generate future carbon mapping schemes on a larger scale, contributing replicable
methodologies based on open data and interpretable models. This possibility is particularly relevant in the context of
European climate neutrality objectives, where the accurate quantification of carbon in forest soils is a priority (Panagos et al.,
2020).

## 4.5 Study limitations and methodological recommendations

Estimating SOC using metrics derived from LiDAR and forest inventory data offers an innovative and non-destructive
approach to assessing carbon reservoirs in forest ecosystems. However, this study has certain limitations that should be
considered when interpreting the results and planning future research. One limitation is related to the resolution and coverage
of the LiDAR data used, which were obtained from the PNOA. Although these data are publicly available and cover large
areas, their point density and temporal resolution may be limited, which affects the accuracy of the detailed characterization





of forest structure and, consequently, the estimation of SOC (Johnson et al., 2022). This restriction has direct implications for the generalizability of the predictive models developed. Furthermore, although relevant variables such as bulk density and percentage of coarse elements were incorporated, other important edaphic factors such as soil texture, pH or indicators of biological activity were not included, variables that in other studies have been shown to be determinants for the accuracy of SOC estimation (Muñoz-Rojas et al., 2015).


In order to improve future methodological approaches, the integration of multiple data sources is recommended. Combining LiDAR metrics with information obtained from hyperspectral images or radar data would allow for the capture of a wider range of variables related to both vegetation and soil properties, thus strengthening the robustness of the models (Tafur et al., 2022). It is also advisable to increase the size of the field sample, increasing the number of plots and their spatial 535 distribution, to better represent the environmental heterogeneity of the study area and reduce biases in modelling. The incorporation of additional soil variables, such as soil texture, nutrient content or microbial biomass, could also enrich the understanding of the processes that control carbon storage in the soil. Another important limitation is the spatial and temporal scale of the study. Although the approach applied has shown good results in a relatively homogeneous area, its applicability to more heterogeneous landscapes with marked variations in topography, land use or climatic conditions may 540 be compromised. Furthermore, the use of point data limits the possibility of analyzing the evolution of SOC over time. The integration of multi-temporal or repeated data would not only improve the accuracy of the models but also allow the stability of edaphic carbon in the face of disturbances to be assessed (Guillaume et al., 2021). Finally, it is important to implement rigorous cross-validation procedures and use independent test sets to assess the stability and generalizability of predictive models under different forest scenarios.

**5.      Conclusions**

This study confirms the potential of airborne LiDAR data, combined with estimates of aboveground and belowground biomass and soil data, for predicting SOC content in *Pinus halepensis* forest ecosystems. The Random Forest model showed the best predictive performance, achieving a coefficient of determination ($R^2$) of 0.811 and moderate mean errors (RMSE = 7.73 Mg/ha; MAE = 6.13 Mg/ha) during cross-validation, clearly outperforming the parametric models used. The integration 550 of structural metrics derived from publicly available LiDAR data allowed the spatial variability of SOC to be considered efficiently, without the need for intensive field sampling. These results are consistent with previous studies that support the use of machine learning models in the prediction of complex soil properties, especially in systems with high structural heterogeneity. However, limitations related to LiDAR data resolution and the absence of certain key soil variables that could improve our results. Our results reinforce the usefulness of remote sensing and machine learning tools for estimating carbon 555 stocks in forest soils, providing a methodological approach to support sustainable forest management and environmental monitoring strategies in Mediterranean contexts.



**CRediT authorship contribution statement**

**DMP**: writing —original draft preparation, review and editing, conceptualization, methodology, data curation and formal analysis, supervision. **MBT**: writing—review and editing, methodology, supervision. **FBO**: writing—review and editing, methodology, supervision. **IRB**: methodology, supervision. **CHA**: writing—review and editing, conceptualization, data curation and formal analysis, supervision. **FTS**: writing—review and editing, conceptualization, data curation and formal analysis, supervision.

**Declaration of Competing Interest**

We have nothing to declare.

**Acknowledgements**

This study was made possible thanks to funding from the IMFLEX project (PID2021-1262750B-C22) funded by Agencia estatal de investigación, fondos FEDER, Micinn. Ministerio de Ciencia e innovación. Plan de recuperación, transformación y resiliencia, Unión Europea. I would like to thank my tutors, Celia and Frederico, for their invaluable help and advice, as well as my colleagues, especially Rubén de Prado, for his help in collecting field samples, and Marina Ortiz and Elisa Pérez for their assistance in the laboratory. Of course, I would also like to thank my family, especially my partner, for their constant support over the last few months.

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
