# Peer review of "Estimating soil organic carbon stocks in *Pinus halepensis* mill. stands using lidar data and field inventory"

_EGUsphere, 2025_

## Referee Comment (RC2)

**Comments on estimating soil organic carbon stocks in Pinus halepensis mill. Stands using LiDAR data and field inventory**

This manuscript presents a study on estimating soil organic carbon (SOC) stocks in *Pinus halepensis* stands based on the random forest algorithm by integrating airborne LiDAR metrics with field forest inventory and soil data. The topic is relevant for upscaling SOC estimates using remote sensing.

The paper is well written, but it places more emphasis on the technical descriptions with little ecological indications. For example, why should canopy height variability be related to SOC?

There are some limitations of this study. For instance, a baseline model against which the added value for LiDAR data is missing. The manuscript presents models that include LiDAR metrics, but does not evaluate SOC predictions based on field soil variables or forest inventory data alone. Consequently, it remains unclear whether LiDAR substantially improves SOC estimation, or whether similar performance could be achieved without remote-sensing inputs. Secondly, the models are validated only through cross-validation but not on an independent dataset, which hampers confidence in generalizability. Thirdly, LiDAR data could provide information on the substrate for forming SOC, but edaphic factors especially soil texture, pH and microbial biomass are crucial factors for SOC. By not explicitly quantifying the contribution of these edaphic controls, the manuscript cannot disentangle whether LiDAR explains SOC directly, or merely acts as a proxy for site conditions. Last but not least, the modeling workflow is generally well structured and follows good practices, but several aspects would benefit from clarification to improve transparency and reproducibility. Detailed comments please see below:

1) The author mentioned 97 structural metrics were found but did not mention if there was any feature selection or dimensionality reduction applied to avoid overfitting.

2) The Introduction part the author mentioned that "there are not many studies about modelling SOC using LiDAR metrics" (line 127), this gap could be articulated more forcefully. The novelty relative to existing work isn't sufficiently clear. For instance, compared to the previous studies, what's different? More structural metrics? Better validation? A different study area or tree species?

3) Lines 151-155 identify "scale mismatches, spatial resolution, and model generalisability" as challenges but don't explain how this study addresses them. Does integrating these variables actually solve the problem, or is it acknowledged as a limitation? This needs clarification.

4) Please discuss why ALS metrics are effective predictors and how these relate mechanistically to SOC process?

5) In Table 5 it shows the increasing trend of organic carbon in the topsoil with bulk density, which I was suspicious, because generally, when SOC increases, the soil becomes looser and less compact, lowering bulk density. Please clarify this.

6) In Table 7 it shows that the logarithmic model has the highest $R^2$ while RF has the lowest RMSE and MAE, but it does not explain why these metrics diverge, and what this implies about prediction accuracy versus variance explained. Please could the author add a brief interpretation of this discrepancy?

7) In Figure 4 please add descriptions on OM, BD CD, C data into the caption.

8) Please use the lowercase of i in the Equation 2. Please also give the info of the horizon thickness (m) used in this study, so that the audience gets info on measurement depths.

9) Please move (SAS Institute Inc., 2023) behind the SAS software.

10) In line 259, it is better to add the information of the percentage of data availability.

11) In line 260-261, could you please clarify why you use the criterion of the heights of less than 2 m as terrain? Do you validate against your ground truth?

12) In line 263, could you briefly specify the tree segmentation algorithm and parameters used?

13) In line 265, could you clarify the reason why you use the IDW method rather than the TIN-based method for generating DTM?

14) In line 269, could you please briefly describe how negative normalized heights (e.g., due to interpolation artifacts) were handled and specify the spatial resolution of the DTM in this study?

15) In line 270, could you please indicate the spatial resolution of the CHM, because this greatly affects tree crown detection accuracy.

16) In line 286, please specify how LAD metrics were derived (e.g., which algorithm or model was used to estimate LAD from LiDAR returns), as different methods can yield substantially different results.

17) In line 288, the author mentioned all metrics are used as predictors, please briefly describe if there are redundancy and multicollinearity and how the author handles this?

18) In line 280-289, please add citations for commonly used metric families (percentiles, LAD, L-moments).

19) In line 305, does the author mean that only mtry and min_n were tuned, how about other hyperparameters (e.g., number of trees, node depth, sampling scheme), which can also influence model performance. Please clarify whether these values were kept at defaults or fixed manually.

20) In line 365 'total carbon tree biomass' is unclear. Does root biomass account for a portion of total carbon biomass, or are you implying that root biomass equals total carbon biomass? Rephrase for accuracy.

21) In Table 7 please add units for RMSE and MAE.